# G-Protein-Coupled Receptors in CNS: A Potential Therapeutic Target for Intervention in Neurodegenerative Disorders and Associated Cognitive Deficits

**DOI:** 10.3390/cells9020506

**Published:** 2020-02-23

**Authors:** Shofiul Azam, Md. Ezazul Haque, Md. Jakaria, Song-Hee Jo, In-Su Kim, Dong-Kug Choi

**Affiliations:** 1Department of Applied Life Science & Integrated Bioscience, Graduate School, Konkuk University, Chungju 27478, Korea; shofiul_azam@hotmail.com (S.A.); mdezazulhaque@yahoo.com (M.E.H.); md.jakaria@florey.edu.au (M.J.); wowsong333@naver.com (S.-H.J.); 2The Florey Institute of Neuroscience and Mental Health, The University of Melbourne, Parkville, VIC 3010, Australia; 3Department of Integrated Bioscience & Biotechnology, College of Biomedical and Health Science, and Research Institute of Inflammatory Disease (RID), Konkuk University, Chungju 27478, Korea

**Keywords:** serotonin, cannabinoid receptor, metabotropic glutamate receptor, seven transmembranes, orphan G-protein-coupled receptors

## Abstract

Neurodegenerative diseases are a large group of neurological disorders with diverse etiological and pathological phenomena. However, current therapeutics rely mostly on symptomatic relief while failing to target the underlying disease pathobiology. G-protein-coupled receptors (GPCRs) are one of the most frequently targeted receptors for developing novel therapeutics for central nervous system (CNS) disorders. Many currently available antipsychotic therapeutics also act as either antagonists or agonists of different GPCRs. Therefore, GPCR-based drug development is spreading widely to regulate neurodegeneration and associated cognitive deficits through the modulation of canonical and noncanonical signals. Here, GPCRs’ role in the pathophysiology of different neurodegenerative disease progressions and cognitive deficits has been highlighted, and an emphasis has been placed on the current pharmacological developments with GPCRs to provide an insight into a potential therapeutic target in the treatment of neurodegeneration.

## 1. Introduction

G-protein-coupled receptors (GPCRs) constitute the largest family of membrane receptors in humans, and ~34% of marketed drugs target this family. Since their discovery, this receptor family has been both pharmacologically and biologically important in the treatment of different pathological conditions. GPCRs contain a common pattern in their basic structures. They consist of seven membrane-spanning domain structures, which they use to sense different signals and stimuli, including light, stress, hormones, peptides, and so forth [1].

Neurodegenerative diseases (NDDs) are most common and prevalent in elderly people worldwide [2] and cause progressive neuronal dysfunction, toxicities, and death. These diseases lead to an irreversible weakening of all brain functions. Around 30 million individuals worldwide are affected by NDDs [3,4]. Alzheimer’s disease (AD), vascular dementia (VaD), frontotemporal dementia (FTD), Parkinson’s disease (PD), and Huntington’s disease (HD) [5] are the prevalent NDDs commonly diagnosed in aged people. The cognitive function in patients is seriously affected during these disease’s progression. Therefore, the age-related decline in cognitive function is a leading challenge in mental health research [6]. Although some symptomatic treatments are available for NDDs, no specific treatments have yet been found [7]. Among the various NDDs, AD, which causes impaired cognitive function and memory loss, is the most extensively discussed and researched [8]. Amyloid β (Aβ) plaque formation and the hyperphosphorylation of tau proteins in the neurofibrillary tangles of the frontal and temporal lobe and other regions of the neocortex are observed during the progression of AD. VaD is another common cause of dementia, in which patients often complain of disturbances in frontal executive function [1] and multiple cerebrovascular pathologies, including vessel occlusion, arteriosclerosis, hypertension, aneurysms, and various forms of arteritis [8]. FTD can be diagnosed in people less than 65 years old [9] and is characterised by neuropsychiatric symptoms and behavioural, motor, and cognitive impairments [10]. Abnormal depositions of tau proteins are observed in different regions of an FTD patient’s brain, such as the hippocampus, frontal cortex, and striatum [11].

Extensive research is ongoing in an attempt to clearly understand the pathobiologies of these NDDs. Within this context, several pieces of evidence strongly suggest that GPCRs are involved in the pathogenesis of several NDDs, including AD, PD, HD, and MS [8,12]. Around 90% of the ~370 nonsensory GPCRs have been found to express in the brain, and they play important roles in regulating mood, appetite, pain, vision, immune responses, cognition, and synaptic transmissions [8]. GPCRs are now the most common target used to develop novel therapeutics. In this article, we present GPCRs and their involvement in the pathophysiology of NDDs and discuss the changes in GPCR expression in different neurodegenerative conditions in light of the recent research in that field. The aim is to provide a potential and novel target for the development of novel therapeutics for NDD treatment.

## 2. GPCRs’ Relevance in Cognition and Neurodegenerative Disorders

GPCRs comprise the largest family of transmembrane receptors, with over 800 members present in humans [13,14]. Slightly over half of the 800 receptors have sensory functions that mediate olfaction (400), taste (33), light perception (10), and pheromone signalling (5) [15]. More than 90% (370) of the remaining receptors are expressed in the brain. These nonsensory GPCRs mediate signalling from multiple ligand types and regulate several physiological processes throughout the human organism, mainly endocrine and neurological processes [8,13,16] (Figure 1). GPCRs are currently the most common therapeutic drug target [13]. In 2017, Hauser and coworkers accumulated data from the Center Watch Drugs in Clinical Trials database and, by cross-referencing with public sources, reported 475 approved drugs that target GPCRs, representing 34% of all FDA-approved drugs [13,17]. This report also indicates that about 20% to 50% of drugs have GPCRs as a target, with discrepancies probably resulting from the varying definitions of ‘drug target’ [16,18,19]. As new functions for GPCRs are discovered, especially in the case of the 100 orphan GPCRs for which no endogenous ligand or clearly defined function are currently known, the number of drugs targeting GPCRs is expected to increase [14,19]. Therefore, the relevance of the different types of GPCRs in neurodegeneration and age-related disease progression is the focus of this article.

### 2.1. Alzheimer’s Disease

AD is the predominant form of NDD and is most prevalent among people over the age of 65. AD patients exhibit cognitive deficits, memory loss, and changes in personality and behaviour [8,22]. The pathophysiology includes an accumulation of amyloid β (Aβ) plaques and the aggregation of tau proteins in neurofibrillary tangles, which are detected primarily in the frontal and temporal lobes and later slowly progress into other regions of the neocortex [22]. With the progression of this disease, different regions of the brain degenerate, and their neurochemical pathways change, including acetylcholine, serotonin and adenosine homeostasis, leading to cognitive impairments.

Although AD treatment strategies are progressing, they are limited to symptomatic relief only. One of the common features of AD is the gradual downregulation of the acetylcholine level in the brain, which is due to the abundance of acetylcholinesterase [23]. Acetylcholinesterase inhibitors (AChEi), therefore, play a vital role in the symptomatic treatment of AD and also improve cholinergic neurotransmission and cognitive function [24]. However, glutamatergic neurotransmitter overexpression is also reported in the pathogenesis of AD [25,26]. In addition, glutamate receptors are reported to mediate most of the excitatory neurotransmissions in the mammalian brain, and their overstimulation can cause excitotoxicity [25,27]. Metabotropic glutamate receptors (especially mGluR5) have been found to be involved in both cognition and Aβ generation and accumulation. A study showed that genetic deletion of mGluR5 in the mouse (APPswe/PS1ΔE9) ameliorates cognitive deficiencies and minimises Aβ production [28]. The study also indicates that mGluR5 expression is related to the increase in Aβ plaque and mediated by the mechanism of the fragile X mental retardation protein (FMRP) [28]. However, the mGluR5 antagonist 3-[(2-methyl-1,3-thiazol-4-yl) ethynyl]-pyridine (MTEP) has efficiently reversed this condition under the same paradigm [29]. Extensive evidence indicates that a wide-ranging serotonergic denervation of the neocortex and hippocampus occurs in AD patients. For example, activation of the serotonergic 5-HT_2A_ and 5-HT_4_ receptors improves learning and memory [30,31] through the activation of the extracellular signal-regulated kinase (ERK) mediated by either the G-protein or β-arrestin (Figure 1) [20]. Therefore, targeting serotonergic receptors would also intervene in AD progression.

### 2.2. Vascular Dementia

Vascular dementia (VaD) is predominant in 8–15% of cognitively-impaired aged patients. Neuropathological investigation of VaD shows multifocal and/or diffuse lesions in the subcortical and strategically important brain areas, including the thalamus, frontobasal, and limbic system, and hippocampal sclerosis to multi-infarct encephalopathy [32]. These pathophysiologies further impair cognition, behaviour, execution, and memory. Further, mAChR1 reduction due to cerebrovascular occlusion results in cognitive dysfunction in VaD [33]. In addition, D1R reduction in the hippocampal dentate gyrus in a VaD rat model resulted in learning and memory deficits [34]. The γ-aminobutyric acid receptor (GABAR) in the dentate gyrus regulates synaptic plasticity, learning, and memory. In a VaD rat hippocampus, both GABAR1 and GABAR2 were found to be reduced, which resulted in spatial learning and memory deficits [35].

The 5-hydroxytryptamine receptors (5-HTRs) are prevalent in the frontal and temporal cortices [36] and are key players in cognitive function and memory formation. In particular, the 5-HT_1A_ and 5-HT_2A_ receptors are related to the correlation and preservation of cognition [37]. Reduced 5-HT_1A_ receptors in AD brains leads to cognitive impairments [8,38] and could also affect VaD as well. Hence, although GPCRs have not yet been studied extensively in VaD models, GPCRs could be potential therapeutic targets in VaD as well.

### 2.3. Frontotemporal Dementia

Frontotemporal dementia (FTD) is a special type of dementia affecting both the front and sides of the brain. FTD causes verbal, executive, and behavioural impairments. FTD is mostly diagnosed in people younger than 65 [9], with the disease progression starting at ~45 years of age. As in other types of dementia, the patient’s condition worsens with age. FTD, in some cases, also has a microtubule-associated-protein-tau-based pathology. The mutations in the tau gene lead to different forms of tauopathy [39], including progressive supranuclear palsy and corticobasal degeneration, although most FTD patients do not show apparent tau gene or protein anomalies [39]. Clinical management of FTD is quite challenging, and it is important to understand the neurobiological substrate in order to ease the process. The role of GPCRs in FTD-induced cognitive deficits makes them an interesting target in terms of halting the disease’s progression.

FTD is a combination of impairments in neuropsychiatry and social behaviour. A recent study demonstrated that the neuropeptide oxytocin impacts social cognition and behaviour in FTD patients [40]. Oxytocin and vasopressin, another neuropeptide, are important in social cognition, trust, behaviour, and facial expressions. These neuropeptides are related to the pituitary and synthesised in the hypothalamus, primarily in large magnocellular neurons located in the supraoptic and paraventricular nuclei [41]. They are released into various brain regions, such as the amygdala and anterior cingulate cortex, for the paracrine signalling of the oxytocin receptor (OXTR) and vasopressin receptor (AVPR1a), and these receptors have a pathophysiological impact on FTD [42]. Intranasal vasopressin administration improves social cognition and facial expressions in humans, especially in the eye region [42]. Oxytocin administration in a similar paradigm also improved facial expressions and social interactions [42]. The presence of oxytocin also facilitates GABAergic synapse development, which blocks fear and anxiety [8,43].

Several serotonin receptors, including 5-HT_1A_ and 5-HT_2A_, decline in the anterior cingulate cortex [44], orbitofrontal and medial prefrontal cortexes [45], and the frontal and temporal cortexes [44] of FTD patients. However, in AD patients, the levels of the 5-HT_1A_ and 5-HT_2A_ receptors fall in the hippocampus and prefrontal cortex region, which causes cognitive deficits. In contrast, under similar conditions, the FTD patient develops neuropsychiatric symptoms [38]. Moreover, the mGluR5 and NMDA receptors are colocalized in cortical regions of the brain and are also involved in the formation of receptor-dependent synaptic plasticity [46]. A recent study showed mGluR5 decreases in the paralimbic cortex region of FTD patients and may lead to the neurodegeneration observed [47]. Finally, mGluR5 activation enhances NMDA receptor action, while inhibition of the receptor blocks NMDA receptors [48].

### 2.4. Parkinson’s Disease

Parkinson’s disease (PD) is another prevalent neurodegenerative disease and is the second most ubiquitous age-linked disorder. The aetiology of PD includes motor and nonmotor dysfunctions [49,50,51]. One-quarter of PD patients are reported to develop mild cognitive impairment (MCI), with declines in memory (13.3%), visuospatial (11.0%), and attention/executive function (10.1%) [52]. A 20-year follow-up study of some newly diagnosed PD patients reported the occurrence of dementia in 83% of the cases [53]. However, most PD patients are diagnosed with elevated dopamine D1 and D2 receptors in the striatum and substantia nigra, which leads to dopamine denervation supersensitivity [54]. A study of PD animals showed that a D1 agonist improved visuospatial accuracy, whereas the use of a D2 antagonist induces attention impairment [55]. Therefore, the crucial roles of D1 and D2 in visuospatial and attentional dysfunction in PD–MCI are well supported. A good number of nondopaminergic transmitters are also affected in PD patients, and these transmitters are considered to contribute to cognitive deficits [56]. A post-mortem brain study of PD reported the loss of the cholinergic system, leading to cognitive decline. The cholinergic deficit is also associated with late-onset dementia in PD, whereas Aβ plaque and Lewy bodies are associated with an earlier onset of dementia [57,58]. Several serotonergic receptors are also found to be associated with cognition PD, such as the 5-HT_1B_ receptor, which has been found to be downregulated in the early stage of PD progression [59].

### 2.5. Multiple Sclerosis

Within the context of multiple sclerosis’ (MS’s) pathophysiology, around 40–60% of patients complain of cognitive dysfunction [60]. Previously, it was supposed that depression and cognition are two independent domains that persist in MS. Eventually, it was understood that at a certain point in MS-induced depression, MS patients experience severe difficulties with working memory, execution, and information-processing time [61,62,63]. When proinflammatory cytokines in the central nervous system (CNS) increase, less serotonin is released from the synapses, causing the depression and resulting in the malfunction of the noradrenergic and serotonergic circuits. In this context, antidepressant therapeutics alone might not be sufficient to improve cognitive function in MS. For example, a study conducted in 2008 with two groups of MS patients reported only a slight difference between the group being treated with antidepressants and the placebo group. In that study, 57% of MS patients treated with paroxetine (a selective serotonin reuptake inhibitor (SSRI)) received 50% lower depression scores on the Hamilton Rating Scale, while the placebo group reduced their scores by 40% under similar conditions [64].

### 2.6. Huntington’s Disease

Progressive motor, behavioural, and cognitive impairment are the common features of Huntington’s disease (HD). Cognitive impairment is inevitable in HD. A great deal of evidence indicates that glutamate-mediated excitotoxicity, which includes wide mGluR2 and mGluR5 expression in the neocortical layers, hippocampus, striatum, thalamus/hypothalamus, and cerebellum, is one of the major contributors to HD progression [62,65,66]. It is also evident that mGluR regulates motor function and Huntington (HTT) protein aggregation in HD. A study showed that treatment with the mGluR5 antagonist 2-methyl-6-(phenylethynyl)-pyridine (MPEP) reduces hyperactivity and improves motor coordination in R6/2 HD transgenic mice [67]. Genetic deletion of mGluR5 has also been reported to improve motor function and reduce HTT protein aggregation in HdhQ111/Q111 knock-in mice [67]. On the other hand, both D1R and D2R expression declines in the early and late phase of HD [68]. The hyperactivity of dopamine in the early phase of HD, i.e., activation of D2R, increases HTT protein aggregation [69]. Thus, suppressing/inhibiting D2R with an antagonist would decrease HTT protein aggregation and reduce cell death via striatal protection, which may be beneficial in the treatment of early-stage HD and could potentially delay disease progression [70].

## 3. Therapeutic Targets for Neurodegenerative Disorders Based on Different GPCR Classes

Due to the selectivity in the function, specific distribution, and number of family members, GPCRs possess a potential therapeutic index and are important targets in different neurological conditions; some of the GPCRs listed in Table 1 are therapeutic targets for cognition and memory deficits. The important roles of different GPCRs are discussed below.

### 3.1. 5-HT Receptors

The serotonergic neurotransmitters play a critical role in cognitive and behavioural functions. These receptors are distributed in different brain regions, including the cortex, amygdala, and hippocampus, and are associated with learning and memory [71]. Their roles in cognition and memory have made them important drug targets in neurodegenerative disorders. Some evidence suggests that the use of agonists of the 5-HT subtype, such as 5-HT_2A/2C_ or 5-HT_4_, can prevent memory impairment and improve learning ability. A similar effect can also be obtained by using an antagonist of 5-HT_1A_ or 5-HT_3_ and the 5-HT_1B_ receptor [72].

The 5-HT_6_ receptor (5-HT_6_R) has been shown to regulate several neurotransmitter pathways, including the serotonergic, cholinergic, glutamatergic, and GABAergic systems [73]. In addition, several studies have established that this receptor is involved in learning and memory processes. These studies also claim that 5-HT_6_R antagonists could improve cognitive functions; some agents are in the preclinical stage [74,75]. Many of these agents are associated with substantial improvements in different cognitive tasks and enhanced memory retention or formation in rodents [76]. To date, at least three candidates have already reached Phase II/III clinical trials as novel therapeutic agents for the treatment of AD [77]. Indeed, the serotonin receptor 5-HT_6_R is an attractive drug target for reversing memory loss and learning disabilities associated with NDDs [77,78,79]. A recent study has discovered a new benzimidazole-based compound that is an antagonist of 5-HT_6_R and improves the novel object identification task in memory-deficit mice [76].

Moreover, in a Phase I trial, PRX-07034, a selective 5-HT_6_R antagonist, showed high selectivity for 5-HT_6_Rs over other 5-HT receptors and nonserotonin receptors. This derivative at doses of 1 and 3 mg/kg enhanced short-term memory and improved cognitive flexibility in rats [80]. Although Phase II trials have not yet been concluded [81,82], this drug candidate could be useful in treating dementia in AD [83]. Another selective antagonist, AVN-322, which is a derivative of AVN-101 and AVN-211, is undergoing Phase I trials for AD and schizophrenia. This drug has been reported to reverse the detrimental cognitive effects of scopolamine and MK-80 [84]. The AVN-101 is a multitarget serotonin antagonist that blocks 5-HT_7_R and has a low, but potential, affinity towards 5-HT_6_, 5-HT_2A_, 5-HT_2C_, and adrenergic receptors. It is also a dynamic candidate that is an antagonist of both 5-HT_6_ and 5-HT_7_. Both receptors follow the same signal transduction pathway, which is important for learning, memory, and anxiety. After completing Phase I trials, the drug is poised to start Phase II trials for AD and anxiety [85]. In addition, the benefit of AVN-101 in treating AD progression is the symptomatic relief of anxiety, depression, sleep disorders, and associated mood disorders [85]. Another derivative, AVN-211, has been reported to be a well-tolerated antagonist at a dose of 20 mg/kg of body weight and is suggested in AD therapy for its positive impact on cognition [86].

Idalopirdine is an antagonist of 5-HT_6_R that was developed initially for schizophrenia. Later analysis revealed its positive effect on cognition, and it was recommended for use in AD-associated dementia. Idalopirdine blocks 5-HT_6_R and increases acetylcholine in the CNS. In particular, it increases acetylcholine by inhibiting CYP206, an enzyme that is involved in the metabolism of donepezil. Therefore, cotreatment with donepezil increases donepezil’s bioavailability [87]. During a trial, 90 mg of idalopirdine taken daily with donepezil (10 mg) improved cognitive function significantly [87]. An AChE inhibitor, donepezil, and an NMDAR inhibitor, memantine, were used in a clinical trial involving VaD patients and resulted in cognitive improvement [88]. Monotherapy with intepirdine was found to be well tolerated, but its efficacy was lower than expected in AD. Further, the results of different studies were inconsistent. For instance, one study found a significant change in global function but not cognition [89], while another report failed to find any significant improvements [90,91].

Latrepirdine, or Dimebon, is an antihistamine developed originally for allergic rhinitis. Later, the drug was found to have procognitive effects, which are the result of antagonising 5-HT_6_R. Latrepirdine was also reported to stabilise mitochondria and works as a neuroprotective drug [82]. Based on fact and figures, latrepirdine was recommended for clinical trials in both AD and HD in the 2000s [82]. After completion of a small pilot study, the drug passed its Phase I and II clinical trials. However, in the Phase III trial, use of latrepirdine was terminated due to lack of efficacy [82,92]. SUVN-502 is another selective 5-HT_6_R antagonist. After completing Phase I trials with no notable adverse effects and better tolerance [93], the drug is currently in Phase II clinical trials involving patients with moderate AD [82]. This drug has been reported to promote acetylcholine and glutamate in the CNS when administered in combination with donepezil [94,95]. Agomelatine, a 5-HT_2C_ antagonist, is another drug undergoing clinical trials that promises to improve prefrontal dopaminergic tone in FTD patients [96]. One clinical trial involving agomelatine reported that it reduces depression in AD patients significantly and decreases depression and motor symptoms in PD patients [97,98]. These data suggest that the 5-HT_2C_ receptor could also be an important target in neurodegenerative disorder therapy.

Tryptophan (TRP) is an essential amino acid and serotonin precursor, and its depletion can disrupt serotonin synthesis in the brain. That depletion could also produce a paradigm of the serotonin deficit models, which has been demonstrated in different animals, including mice [99], rats [100], primates [101], and humans [102]. Several studies have assessed the relation between TRP depletion and cognitive function. For instance, Mendelsohn et al. [103] used a diet lacking in TRP but enriched in large neutral amino acids to produce an almost 50% downregulation of 5-HT levels in the cortex, striatum, and hippocampus [103]. That study, however, did not find any significant effect of TRP depletion on spatial, episodic, or working memory and instead reported semantic memory improvement even in the depleted state [103].

Moreover, despite the modest effect of TRP depletion on sustained attention, such as vigilance, it does not significantly affect specific impaired executive functions such as planning, decision making, and responding [103,104]. To the contrary, tryptophan-rich supplementation of a regular diet can enhance serotonin synthesis, and a clinical study found that tryptophan supplementation improved reaction times, visual memory, and attention [105]. Chronic ingestion of docosahexaenoic acid phospholipids with melatonin and tryptophan for 12 weeks also improves mild cognition impairment in elderly patients [106].

Vortioxetine [13], a serotonin transporter inhibitor (SERT), also works as an antagonist of 5-HT_3_, 5-HT_7_, and 5-HT_1D_ receptors, an agonist of 5-HT_1A_, and a partial agonist of 5-HT_1B_. A recent meta-analysis of a 6–8-week treatment with vortioxetine (5-20 mg/day) produced an incremental reduction in depression symptoms, and an increasing effect was associated with an increase in dose [107]. Vortioxetine was also shown to improve cognitive performance in patients with acute major depressive disorder [107]. Similarly, a study involving chronic citalopram use (20 mg; single dose daily) resulted in improved impulse response and contextual information processing abilities on a delayed nonmatching to sample task (DNMST) in healthy controls (n = 20). Acute administration (24 hr) of that drug showed no effect on working memory or impulsive responses. The authors of the study also suggested that DNMST makes a contribution to the activation of 5-HT_1A_ receptors in the entorhinal cortex and hippocampus [108].

Chronic consumption of a diet high in saturated fats and low in fibre is associated with obesity and cognitive decline. In this context, dietary supplementation can prevent cognitive decline by altering serotoninergic signalling in the brain. A recent study with high-fat, low-fibre (5% dietary fibre)-induced obese models has demonstrated an association with the upregulation of 5-HT_1A_R and 5-HT_2A_R binding density in the rat brain in comparison to the low-fat diet group [109]. With the inclusion of galacto-oligosaccharides and resistant starch, receptor binding densities in the hippocampal and hypothalamic region are reduced, improving cognitive function.

### 3.2. Dopamine Receptors

In the late 1950s, Carlsson identified dopamine as a potential neurotransmitter in the brain. Later, it was discovered that a progressive decrease in dopamine is associated with the pathophysiology of PD [51,110,111]. This finding introduced levodopa, the metabolic precursor of dopamine, into the symptomatic treatment of PD. In the past few years, several investigations have been conducted on dopamine and cognitive function [112,113,114,115]. Although dopamine neurons are very few in number compared to the total neuronal population in the brain (< 1/100,000), they are involved in neuroendocrine regulation, mood, motivation, and psychological processes, including working memory and learning [111,112]. Although no specific mechanism has yet been confirmed, the inhibition of tyrosine hydroxylase activity and tyrosine conversion into dopamine and norepinephrine are involved in long-term memory formation [116]. For example, vasopressin is a hypothalamic neuropeptide found to improve memory and does so by interacting with dopamine in the amygdala and serotonin of the hippocampus [117].

As previously mentioned, some evidence suggests that dopamine modulates working memory, but its specific role is not yet fully defined, and its effect in memory processing has not been clarified. Haloperidol, a D2R antagonist, has recently been investigated for its effect on working memory improvement as well as for its distracting-information-ignoring capability. The study was designed with two testing sessions. In one session, participants took a placebo tablet, and in the other, they took Haloperidol 2.5 mg [118]. The study showed that the deleterious effect of haloperidol on response conflict is associated with the negative effect of the drug on ignoring. The authors also suggest that D2R protects memory content from distraction through a general process, and inhibition of D2R could result in the impairment of response conflict as well as reduced quality of recall [118]. D1R and D2R have been found to be important targets in FTD as well. Both DR antagonists (antipsychotics) and agonists (specifically, D2R) are used frequently in FTD treatment. Commonly used antipsychotics, such as olanzapine, quetiapine, and risperidone, have a high affinity for D2R and dissociate rapidly, resulting in very few side effects [119]. A lack of presynaptic dopaminergic nerve terminal and postsynaptic D2R binding in the striatum is prevalent in FTD patients, and most of them complain of rigidity and bradykinesia. Therefore, FTD patients are currently being treated with the DR agonists carbidopa and levodopa to improve behavioural and psychotic symptoms.

However, SK609, a recently designed selective-small-molecule agonist of D3R, selectively inhibits norepinephrine reuptake as well as increases dopamine by 160% at an i.p. (intraperitoneal) dose of 4 mg/kg. SK609 improved rats’ performance in an attention task. Additionally, SK609 has been reported to improve cognitive function in low-performing rats. Interestingly, the molecule did not produce any side effects mediated by DA transporter (DAT) activity [120], such as spontaneous locomotor activity.

### 3.3. Cannabinoid Receptors

Extensive evidence indicates that the endocannabinoid system has modulatory effects on cognitive function, and these effects have a substantial role in memory acquisition, consolidation, and extinction [121]. However, the introduction of cannabinoid drugs often induces opposite effects, when used in anxiety, cognition, and other behavioural deficiencies, depending on the stress level and the aversiveness of the context [121,122]. It has been demonstrated that the use of an endocannabinoid receptor inhibitor under different environmental conditions has a substantial influence on cognitive function without affecting locomotor or emotional behaviour [122]. Although it is difficult to define the exact role of the cannabinoid receptor, cannabinoid signalling influences memory processing. This influence has been demonstrated in several studies [121,122,123].

In particular, Δ9-tetrahydrocannabinol (THC) is a cannabinoid receptor agonist that acts at CB1R to produce a wide range of biological and behavioural activity. This ligand’s association with cognitive deficits and poor decision making has been identified. Using Wistar rats in a rat gambling task study, it has been reported that a high dose of THC reduces premature responses, while another synthetic agonist produced the opposite reaction [124]. Although acute or limited chronic use of THC does not affect subjects, long-term exposure can impair impulse control and attentional function [125]. Therefore, chronic activation of CB1R or antagonism can impair or improve task performance, which makes it an interesting target. For example, in an animal study, the CB1 antagonist rimonabant produced an improvement in decision making [124]. Meanwhile, cannabinoid receptor agonists, such as AEA (N-arachidonoyl ethanolamine) and noladin, have been reported to protect against Aβ-induced neurotoxicity in the human teratocarcinoma cell line NTERA-2/cl-D1. The effect was exerted through the CB1R- and mitogen-activated protein kinase (MAPK) pathway [126]. Treatment with the CB1R antagonist SR141716A failed to protect against Aβ-induced amnesia [127]. Another study with a triple transgenic mouse model of AD (3xTg-AD) reported that CB1R is up-regulated in the anterior thalamus at the age of 4 months, while the CB1R activity decreased gradually in the nucleus basalis of Meynert at 15 months of age [128].

Activation of CB1R and CB2R was reported to affect the upregulation of PPARγ signalling in an animal model study [129], where Aβ-induced neuroinflammation, neurodegeneration, and spatial memory impairment was attenuated. Activation of CB2R using a lower-dose agonist, JWH-015, eradicated native Aβ from human tissue and cleared a synthetic pathogenic Aβ peptide in a human macrophage cell line (THP-1). This effect was attributed to the antagonist SR144528, which is selective for CB2R, and the plaque removal effect induced by JWH-015 was reversed [130]. A selective CB2R agonist 1-((3-benzyl-3-methyl-2,3-dihydro-1-benzofuran-6-yl) carbonyl) piperidine (MDA7) has shown a modulatory effect on cognitive impairment induced by bilateral microinjection of Aβ (1-40) fibrils into the hippocampal CA1 area of rats. Intraperitoneal treatment of MDA7 (15 mg/kg) for 14 days reduced microglial CD11b expression, promoted Aβ clearance, and restored synaptic plasticity, cognition, and memory [131].

The increased expression of CB2R has been noted with reduced CB1R in late-onset HD, mainly in glial cells [132]. The selective CB2R antagonist, SR144528, provides striatal neuron protection in HD rats; the mechanism includes glial cells [133]. Genetic deletion of CB2R also exacerbates HD, but CB2R-selective agonists can reduce striatal neurodegeneration through microglial activation [134]. SR141716 has also been reported to aggravate malonate-induced striatal pathology in HD rats [135]. However, some studies report that cannabidiol, a phytoconstituent and an allosteric modulator of CB1R [136], can rescue neuronal loss induced by THC by preventing THC-induced CB1R loss [137]. Similarly, VCE-003.2, a cannabigerol derivative, improved the antioxidant barrier in 3-nitropropionic-acid-induced HD mice brains [138]. This agent also showed neuroprotection in SOD1^G93A^ transgenic amyotrophic lateral sclerosis (ALS) mice by targeting CB2R and inhibiting endocannabinoid inactivation [139]. Based on different reports, it is speculated that CBR antagonists may provide a better therapeutic insight into HD treatment.

### 3.4. Cholinergic Receptors

Cholinergic receptors are divided into two classes, including the muscarinic and nicotinic receptor families. These two families are further subclassed according to the occurrence of many ACh receptor subtypes, and their differential dendritic, somatic, axonal, and synaptic localisation contributes to the varied roles that these receptors play in the CNS [140]. Five subtypes of mAChRs (M1 to M5) have been defined and pharmacologically characterized in the CNS. Their expression has been detected at very high levels in the subcortical structures and the cerebral cortex [141]. A modest level of mAChR expression was also reported in the frontal cortex, parietal cortex, temporal cortex, entorhinal cortex, occipital cortex, and insular and cingulate cortexes, with the highest values recorded for the temporal and occipital cortexes [141]. M1 receptors are the most abundantly expressed mAChRs [141].

Nicotinic AChRs (nAChRs) constitute the second cholinergic receptor type. This type of receptor relies on ligand-gated ion channels and contains five subunits, which are assembled into homomeric or heteromeric subunit combinations. The pharmacological and biological characteristics of these receptors are determined by this combination [142]. The nAChR subunits are composed of α4, α6, α7, β2, and β3 subunits and are mostly expressed in the striatum [143,144]. They are expressed on glutamatergic and dopaminergic neuron terminals, GABAergic interneurons, and cholinergic interneurons (ChIs) but are not present in medium spiny neurons (MSNs) [145,146]. However, nAChRs can express in both pre- and post-synaptic neurons. Thus, they can depolarise and increase excitability, causing glutamate, DA, and GABA to release [147,148].

The nAChRs form good targets in the treatment of neurodegeneration. In recent years, several agonists and partial agonists of the α7 subunit have been evaluated in the treatment of NDDs. Daily nicotine administration in a PD animal model produced improvements in motor coordination. That treatment also showed a beneficial effect on neuronal survival, as well as on microglial and astrocytic activation, providing neuroprotection against MPTP/MPP^+^ toxicity [149]. DMXBA (GTS-21) and ABT-107, both α7nAChR agonists, have also produced beneficial effects by attenuating nigrostriatal damage in 6-hydroxydopamine (6-OHDA)-induced rats [51]. Another agonist, PHA 543613, attenuates early-stage HD induced by striatal quinolinic acid lesions. PHA 543613 reduced microglial activation to protect neurons [150]. In another study, α7nAChR-agonist-treated 3xTg-AD mice showed improved cognitive function [151]. However, PNU-282987 has also produced improved motor activity, anxiety, and learning and memory development in the B6C3-Tg mice model for AD [152]. Similarly, two other agonists, AR-R17779 and ABBF (N-[(3R)-1-azabicyclo [2.2.2]oct-3-yl]-7-[2-(methoxy) phenyl]-1-benzofuran-2-carboxamide), are also reported to improve learning and memory [153,154].

ANAVEX2-73, a σ1R (sigma 1 receptor) agonist and muscarinic receptor ligand, has been studied on the Aβ25-35-injected AD mice model [155]. Because Aβ25-35 injection modulates mitochondrial respiration in the hippocampus, ANAVEX2-73 (0.01-1 mg/kg IP) restored respiration to normal and prevented Aβ25-35-induced increases in lipid peroxidation levels, Bax/Bcl-2 ratio, and cytochrome c release into the cytosol [155].

### 3.5. Metabotropic Glutamate Receptors

Metabotropic glutamate receptors (mGluRs) are members of the C GPCR family, and they consist of eight subtypes (mGlu1 to mGlu8) that are further subdivided into three groups depending on their amino acid sequences, G-protein coupling, and pharmacological characteristics. Group I consists of mGlu1 and mGlu5 receptors, which are coupled to G_q_/G_11_ [156]. The Group II (mGlu2, mGlu3) and Group III (mGlu4, mGlu6, mGlu7, mGlu8) subtypes are coupled to G_i_/G_o_. Both of these groups regulate adenylate cyclase negatively and can also activate the MAP kinase and PI-3-kinase pathways [156,157]. However, overexpression of mGlu5 has been reported in different neurodegenerative disorders [157]. Specifically, Aβ plaques have been found in the surroundings of astrocytes, spinal cord lesions, and MS lesions, ALS, PD, and in the hippocampal astrocytes of Down syndrome patients [158,159].

Several mGluRs agonists, antagonists, and positive and negative allosteric modulators have been studied using different neurodegenerative animal models. Collectively, these data on mGluRs provide an insight into the development of therapeutics for treating NDDs. However, LY341495 (an antagonist to Group I/II mGluR) was reported as blocking Aβ-enhanced long-term depression and improving synaptic plasticity [160]. The same study also reported that pretreatment with an mGluR1/5 agonist, 3,5-dihydroxyphenylglycine (DHPG), decreased Aβ-enhanced long-term depression [160]. SIB1757, a noncompetitive antagonist of mGluR5, prevented Aβ oligomer-induced synaptic N-Methyl-D-aspartic acid receptor NMDAR reduction [161]. A comparison study targeting Group II mGluRs showed that an mGlu2R positive allosteric modulator, N-4′-cyano-biphenyl-3-yl)-N-(3-pyridinylmethyl)-ethanesulfonamide hydrochloride (LY566332), amplified Aβ-induced neurodegeneration [162]. Treatment with the antagonist (2S,1′S,2′S)-2-(9-xanthylmethyl)-2-(2′-carboxycyclopropyl) glycine (LY341495) of mGlu2/3R prevented this effect. Similarly, the dual mGlu2/3 receptor agonist (−)-2-oxa-4-aminobicyclo [3.1.0] exhane-4,6-dicarboxylic acid (LY379268) exhibited neuroprotection via a paracrine mechanism mediated by transforming growth factor-β1 [162]. Therefore, dual activation of mGlu2R and mGlu3R may be a prime target for providing neuroprotection against Aβ-induced toxicity, and negative modulation of mGlu5R would also be a good target for PD and AD treatment.

### 3.6. Orphan GPCRs

More than 140 GPCRs remain mysterious and are referred to as ‘orphans’ (oGPCRs); most of them do not have any known ligands. Although very little is known about their endogenous or exogenous ligands, these so-called oGPCRS have gained considerable attention as drug targets [163]. Several findings have postulated the critical role of oGPCRs in the cognitive deficits in disorders such as AD and schizophrenia [164,165,166,167]. For example, an expression map of the mouse brain has listed 78 oGPCRs and showed that many of them are relevant to cognition, motivation, and emotional processing [168]. That study reported that oGPCRs, such as GPR17, GPR27, GPR37, GPR39, GPR63, GPR85, GPR88, GPR123 (Adgra1), GPR125 (Adgra3), GPR153, GPR176, and GPRc5c, are highly expressed in the prefrontal cortex (PFC) region of the mouse brain, which is involved in cognition and learning. Both mouse and human brain analyses have shown that four oGPCRs are highly expressed, including GPR88, GPR123, GPR149, and GPR151, but information about these oGPCRs is minimal, with the exception of GPR88 [168].

However, GPR3 has exhibited stable and enhanced expression throughout the ageing process in ten different regions of the healthy human brain, which has been linked to AD pathogenesis in multiple cohort studies [166]. Additionally, GPR3, independent of G-protein coupling, recruits β-arrestin 2 and promotes γ-secretase activity, which increases Aβ precursor protein (APP) cleavage and accelerates Aβ peptide production and accumulation [169]. Thus, deletion of GPR3 in the AD mouse alleviated cognitive impairment and restored memory [166]. Although GPR158 improves memory through transducing osteocalcin and regulating IP3 and BDNF in the CA3 neurons [167], GPR158 overexpression in PFC has a potential role in depressive behaviour, which is reversed by its depletion [170].

GPR52 is a nonodorant GPCR and is colocalized with both D1 and D2 receptors in the basal ganglia neurons. Histological experiments suggest that GPR52 promotes and regulates the *Cre-lox* system and may also modulate cognition and emotion, as it has involvement with both dopaminergic and glutaminergic neurotransmission [171]. Thus, the incorporation of the GPR52 antagonist may potentiate cognitive improvement and exert anxiolytic activity in psychiatric disorders [171]. GPR3 knockout also produced anxiety and depressive behaviour, with no noticeable locomotor impairment under stressful conditions. However, the lack of GPR3 has no preventive action in the learning involved in fear memory in a similar stressful condition in mice [172]. GPR3 also regulates serotonin (5-HT) and dopamine (DA) synthesis and reuptake, which makes it a primary target as well. A study has reported the possibility that serotonin reduction in the frontal cortex and hippocampus causes aggressive behaviours in GPR3 knockout mice [172]. This finding indicates that GPR3 modulates the serotonergic and dopaminergic system, which makes it a potential target in the therapy of AD or schizophrenia.

GPR55 is highly expressed in the pyramidal cells in the hippocampal CA1 and CA3 layers and modulates the synaptic plasticity of pyramidal cells [173]. However, GPR85 is highly expressed in the dentate gyrus region of the hippocampus [174,175] and prominently expresses in the phases of neuronal differentiation in the developing cerebral cortex [176]. This expression suggests a possible role of GPR85 in cognition, and this receptor could become a potential drug target as well.

## 4. An Emerging Paradigm in the Development of Therapeutics for Neurodegenerative Disorders

### 4.1. Allosteric Modulators of GPCRs in the Treatment of Neurodegeneration

Allosteric ligands bind to GPCRs at their endogenous ligand-binding sites. This binding is distinct from the conventional regulation of the downstream GPCR effect due to the interaction between agonists and ligand-binding pockets (Figure 2) [177]. Allosteric ligands provide an opportunity to manipulate the GPCR functions for potential therapeutic benefit. However, their complex actions are challenging for new drug screening and development. Several studies focusing on areas such as biased signalling by allosteric ligands have exploited the interaction mechanisms between allosteric ligands and GPCRs, and learning how these interactions modulate the effects would be beneficial for drug discovery.

Allosteric modulators include ions, ligands, small and large molecules, and protein complexes. They could become favourable pharmaceutical products if developed into low-molecular-weight, nonpeptidic molecules able to cross the blood–brain barrier readily. Allosteric modulators are divided into two major categories based on receptor signalling, i.e., positive allosteric modulators (PAMs) and negative allosteric modulators (NAMs) (Figure 2). They neither activate nor inhibit the receptors, unlike the ligand. They do not bind to the conventional binding site but instead bind to a site that is distinct and highly diverse from the active site. Therefore, PAMs and NAMs could reduce side effects, maintain natural rhythm, and control the potency and efficacy of the drug response [178]. Furthermore, recent progress in neurodegenerative disorder research, including PD, AD, and cerebellar ataxia research, has turned up a potential disease-modifying treatment via allosterism [179,180]. The mAChRs subclasses M1 and M4 are major targets for schizophrenia, AD, and PD [181,182]. Although the M1/M4 agonist xanomeline showed improved cognitive functions in a Phase III clinical trial for schizophrenia, xanomeline is associated with gastrointestinal side effects, and a PAM could possibly be a potential and safe alternative. Several pharmaceutical studies have identified active M1 PAMs in lower animal models, but the safety margin needs to be confirmed [183,184]. M1 PAM MK-7622 was terminated after a Phase IIa/IIb clinical trial. However, several selective M4 PAMs, including LY2033298, VU0152100, VU0152099, and VU0467485, have been evaluated in preclinical models of schizophrenia [185]. A recent crystallization approach to the M1 and M4 receptors provided a structural basis for understanding PAMs [186], which could be a potential target to develop drugs for AD. However, allosterism in the treatment of neurodegenerative disorders depends on the availability of the allosteric modulators. Many potential targets need to be proven or disproven in the neurodegeneration treatment options provided by the GPCR family. PAMs or NAMs of these members could efficiently reduce neurodegenerative progression and reduce therapeutic side effects. For example, allosteric modulators of several members of mGluR showed symptomatic therapeutic potential in a preclinical study of AD. Development of these modulators for clinical study in humans could change the therapeutic strategy for AD [178].

### 4.2. Neuropeptides as a Target to Treat Neurodegenerations

Neuropeptides are small amino acid molecules that can regulate neuronal activity. They possess the ability to modulate different functions such as thermoregulation, reproductive behaviour, food and water intake, and circadian rhythms. However, neuropeptides also have a role in inflammatory responses and pain sensitivity. Several neuropeptides have been identified as being related to neuroinflammation, including the proopiomelanocortin (POMC) gene-derived peptides, neuropeptide Y (NPY), vasoactive intestinal peptide (VIP), somatostatin (SST), calcitonin gene-related peptide (CGRP), and cortistatin (CST). Moreover, a mammalian homolog of the nicotinamide adenine dinucleotide (NAD)-dependent deacetylase sirtuin family member, silent information regulator 1 (SIRT1), is related to AD progression. Studies have demonstrated that SIRT1 decreases ROCK1, a serine/threonine Rho kinase, and regulates β-amyloid metabolism by promoting α-secretase enzyme activity. Therefore, SIRT1 promoters could prevent the formation of β-amyloid oligomers and senile plaques by inducing APP processing in the nonamyloidogenic pathway [187]. Natural derivatives, such as resveratrol, oxyresveratrol, and 2,3,4′,5-tetrahydroxystilbene-2-O-b-D-glucoside (TSG), have been reported to enhance or activate SIRT1 and downregulate neurodegeneration [188]. It has also been observed that SIRT1 overexpression decreases the amyloid plaque burden and improves behavioural phenotypes through the deacetylation of retinoic acid receptor β (RARβ), a transcriptional activator of disintegrin- and metalloproteinase-domain-containing protein 10 (ADAM10), that processes APP through the nonamyloidogenic pathway [189]. SIRT1 expression may deacetylate tau proteins directly, diminish the formation of neurofibrillary tangles, and suppress tau-induced pathology [190].

NPY is present in the hippocampus and produced by γ-aminobutyric acid (GABA)-ergic interneurons. NPY is also distributed in the cerebral cortex, hypothalamus, thalamus, brainstem, and cerebellum and plays a major role in the regulation of learning, memory, feeding, and endocrine secretions. Successive reports indicate that NYP expression is reduced with the progression of AD, PD, and HD [191,192]. NYP could protect neuronal cell death from excessive GluRs activation by reducing the Ca^2+^ influx in the presynaptic nerve terminal through the protein kinase A and p38K pathways [193]. NPY also plays an important role in adult neurogenesis regulation. Spencer et al. (2016) developed a technique to evaluate NYP’s role in neurogenesis by fusing NYP vectors to the brain transport peptide (apolipoprotein B) of an APP-tg mouse model. That study showed that the proliferation of neural precursor cells in the subgranular zone of the hippocampus increased significantly without further differentiation into neurons [194]. NPY-regulated neurogenesis can be observed in the dentate gyrus, caudal subventricular zone (cSVZ), and subcallosal zone via the proliferative effect of Y1 receptors on neuroblasts [195]. Moreover, NYP administration protected neurons from microglia-caused inflammation in the striatum and substantia nigra of 6-OHDA-lesioned rats as a PD model [196]. Thus, NPY could be a potential therapeutic target for preventing neurodegeneration.

Finally, ghrelin, a 28-amino acid peptide that possesses the ability to stimulate growth hormone (GH) release from the pituitary, is highly expressed in the hypothalamus. The ghrelin receptor (GHSR) forms a heterodimer with D1R and D2R and can alter G-protein coupling as well as ligand binding potency [197]. Some evidence suggests that ghrelin exerts neuroprotection against NDDs [197]. Leptin, a cytokine-like 167-amino acid peptide, regulates the production of POMC and NYP at the hypothalamus and can modulate neurogenesis, synaptogenesis, neuronal excitability, and neuroprotection in extrahypothalamic sites [198].

## 5. Conclusions

NDDs get worse day by day. Unfortunately, their therapeutics are disappointing and limited to symptomatic treatment. However, targeting the underlying pathobiology of the NDDs would disrupt disease progression and improve the conditions [199]. Herein, several GPCRs and their agonists or antagonists under screening or approved for AD, PD, MS, or HD, have been discussed. In addition, allosteric modulation of GPCRs or the targeting of neuropeptides would intervene in neurodegeneration and improve associated cognitive deficits. Interestingly, the number of GPCRs found to be involved in the pathophysiology of NDDs is increasing and widening GPCR-based targets. For example, CB2R expression increases, along with a noticeable reduction in CB1R, in late-onset HD. Thus, CB2R would be a potential target for HD therapy. Furthermore, α7nAChR is also a potential target, and several potential agonists have been assessed. Some of them have moved to clinical trials. Several oGPCRs have been found to be associated with the pathophysiological conditions involved in neurodegeneration. Several studies have concluded and consider oGPCRs to be novel targets for solving neuropsychiatric disorders, including anxiety, depression, and cognition in AD, PD, HD, and schizophrenia.

Neuropeptides, such as NYP, GHSR, POMC, CGRP, and SIRT1, would also be interesting targets for improving cognitive functions. Their association with neurodegenerative disease progression has been well established in the last few years. NYP has a potential therapeutic role in the most prevalent neurodegenerative and neuroimmune diseases and counteracts depressive symptoms in NDDs. Stimulating these neuropeptides causes neuronal survival and neuroproliferation by attenuating neuroinflammation and excitotoxicity and inducing autophagy. Similarly, SIRT1 and GHSR protect against neurodegeneration and improve behavioural phenotypes. However, the roles of different neuropeptides in NDDs and cognitive improvement are more complex and not yet fully uncovered. On the other hand, several potential AMs of GPCRs have been mentioned lately for the treatment of neurodegenerative disorders. Although many potential candidates exist, there is still a need to discover and develop potential compounds for treatment. For example, AMs of mGluRs have shown potential in symptomatic relief. In this case, selective PAMs or NAMs of mGluR7, mGluR3, and mGluR8 would be potential targets. The mGluR5 NAM has also exhibited disease-modifying effects in an AD animal model study. Thus, the development of GPCR-allosteric modulators would be an alternative therapeutic option for neurodegeneration and cognitive deficits.

A combinatorial therapeutic approach based on disease progression and variety would also be beneficial. Multiple GPCRs would be targeted to either slow down or stop disease progression. For example, inhibition of both A2AR (adenosine 2A receptor) and the corticotrophin-releasing hormone receptor 1 (CRHR1) by inhibitors may improve cognitive function and reduce depression in AD patients. At the same time, combination therapy could overcome monotherapy-associated side effects. Indeed, an in-depth mechanistic study is required to clarify the interactions between different neurochemical pathways in combinatorial therapy. As such, the combination of L-DOPA and the A2AR antagonist istradefylline targets two neurochemical pathways. Istradefylline reduces the relevant side effects of L-DOPA monotherapy in PD patients. GPR3 regulates both serotonergic and dopaminergic synthesis; thus, targeting this receptor with an agonist would maintain 5-HT and dopamine at normal levels and improve depression and aggressive behaviour. In several neuroinflammation cases, it has been observed that abnormal protein accumulation regulates the pathological changes in the brain and causes mitochondrial dysfunction or oxidative stress. Therefore, neuropeptides, such as vasoactive intestinal peptide and pituitary adenylate cyclase-activating polypeptide, which have been shown to inhibit mitochondrial apoptosis and act on GPCRs, could be an alternative therapeutic option for targeting the GPCRs involved in neuroprotection, thus providing neuronal protection against oxidative stress and inflammation and intervention against NDDs. Collectively, recent and past evidence indicates that targeting either single or multiple GPCRs could potentially intervene against NDDs either by preventing or halting disease progression. Recently discussed neuropeptides and allosteric targets that are directly involved in several aspects of neurodegeneration, such as adhesion GPCRs, could become the next frontier in the development of alternative neurodegenerative treatments [178].

In conclusion, natural active compounds show promising neuroprotective effects in several disease models. The potential of these compounds in terms of acting on GPCRs needs to be studied [5,6,200]. Several recent studies reported that nuclear receptor subfamily 4 group A member 2 [199], NACHT, LRR, and PYD domains containing protein 3 [201], and glutamate receptors [202] have great potential in neuroprotective therapy. The correlation between GPCRs and these target molecules would be a fascinating area of research.

## Figures and Tables

**Figure 1 cells-09-00506-f001:**
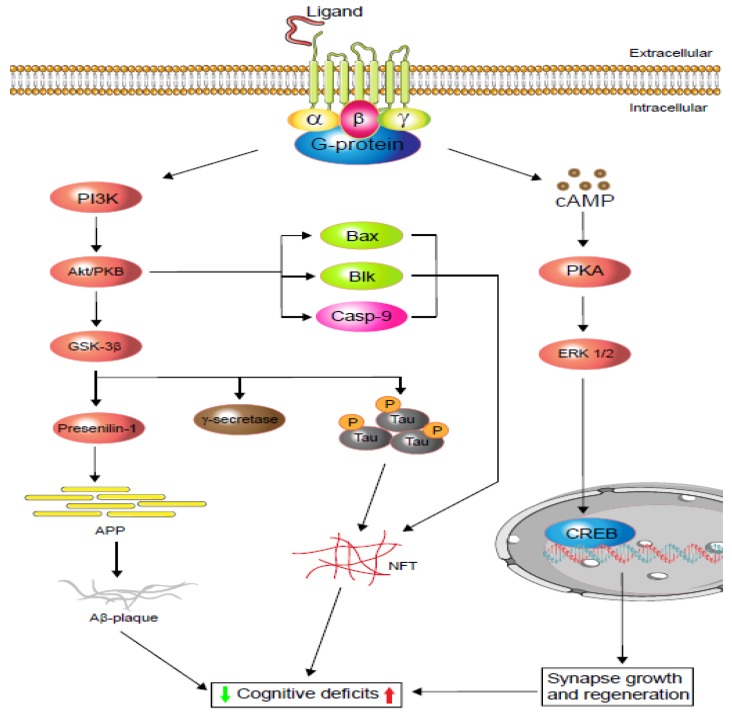
Schematic display of G-protein-coupled receptors (GPCRs) signalling in cognitive impairment. Depending on the agonist or inverse agonist ligand binding, the PI3/Akt-signalling pathway signals Bax and Casp-9 while increasing neurofibrillary tangle (NFT) formation via phosphorylation. PI3/Akt-signalling hyperphosphorylation also activates glycogen synthase kinase-3β (GSK-3β), which increases either or both tau protein phosphorylation and amyloid precursor protein (APP). Phosphorylated tau protein forms neurofibrillary tangles (NFTs) and regulates cognitive function. Similarly, APP metabolism regulates Aβ-plaque formation and controls cognition. Moreover, neuronal and dendritic plasticity is required for synaptic growth, regeneration, and memory formation, and it depends on extracellular signal-regulated kinase (ERK ½) modulation. Depending on the ligands, GPCRs activate cAMP-response element-binding protein (CREB) via the cAMP/ERK ½ pathway and regulate cognition (based on [20,21]).

**Figure 2 cells-09-00506-f002:**
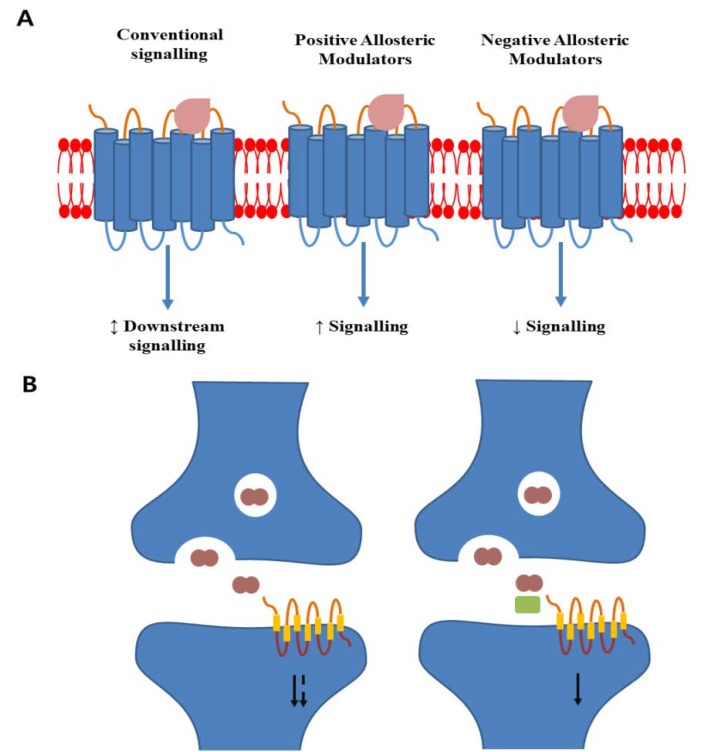
Schematic display of allosteric modulator action on GPCRs. (**A**) Conventional agonist binding makes conformational changes and activates downstream signalling. Positive allosteric modulators bind to a distinct site and enhance conventional ligand-induced signalling. Negative allosteric modulators binding decreases conventional agonist efficacy and reduces downstream signalling. (**B**) In normal physiology, neurotransmitters are released into the synaptic cleft, binding to postsynaptic GPCRs, and activating downstream signalling. The duration of signalling can be degraded by metabolizing enzymes. A positive allosteric modulator (green rectangle) cobinding with the metabolites can extend the duration of receptor activation and enhance signalling (based on [177]).

**Table 1 cells-09-00506-t001:** Therapeutics for different neurodegenerative disorders—new agents, Phase II or III clinical trials [13].

Agents	Targets	Receptor Family	Indications
Erenumab	CALCRL	Calcitonin	Migraine
Ubrogepant	CALCRL	Calcitonin	Migraine
Eptinezumab	CGRP	Calcitonin	Migraine
Cannabidivarin	GPR119	GPR119	MS and epilepsy
Cannabidiol	GPR55	GPR55	MS and epilepsy
VSN16R	GPR18	GPR18	MS and epilepsy
Δ-9-Tetrahydrocannabinol-cannabidiol (THC-CBD)	CB1/2 receptor	Cannabinoid	Spasticity in MS
Fingolimod	Sphingosine 1-phosphate (S1P)	Sphingosine 1-phosphate (S1P) receptor	MS
Xanomeline	M1/M4	Cholinergic	AD
AF267B	M1 and M3	mAChR	Reduces amyloid and tau pathologies in AD
Leuprolide	AChEI	Cholinergic	Synergizes AChEI activities
Vortioxetine	5-HT_3_, 5-HT_7_, 5-HT_1D_	Serotonin	Major Depressive Disorder
Haloperidol	D2R	Dopamine	Working memory in PD
SK609	D3R	Dopamine	Locomotor activity in PD
VCE-003.2	CB2R	Cannabinoid	HD
DMXBA (GTS-21) and ABT-107	α7nAChR	Cholinergic	PD
PHA 543613	α7nAChR	Cholinergic	Early-stage HD
LY341495	Group I/II mGluRs	mGluRs	Improves synaptic plasticity in AD
Galantamine	AChEI	Cholinergic	VaD, AD, PD, and Lewy bodies with dementia
Rivastigmine	AChEI	Cholinergic	VaD, AD, PD, and Lewy bodies with dementia

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
