# Peer review of "G-Protein-Coupled Receptors in CNS: A Potential Therapeutic Target for Intervention in Neurodegenerative Disorders and Associated Cognitive Deficits"

_cells, 2020, doi:10.3390/cells9020506_

Round 1
Reviewer 1 Report
In this review article, Azam et al highlight the role of G protein-coupled receptors (GPCRs) in the pathophysiology of different neurodegenerative diseases and cognitive deficits with an emphasis on the current therapeutic attempts at targeting GPCRs. They state that GPCRs are now the most common target for the development of therapeutics. They comment that the current therapeutic strategies are not very promising and ultimately depend upon symptomatic treatment. In contrast, targeting of the pathobiology of the neurodegenerative diseases would disrupt the disease progression and therefore improve the condition. The authors discuss several GPCRs and their agonists or antagonists that are currently being screened for various neurodegenerative diseases (such as AD, PD, MS or HD).
The authors conclude that some natural active compounds show some promising neuroprotective effects in several disease models and they suggest that these compounds should be tested in combination with GPCRs.
Comments:
This is quite a nice review that covers the main neurodegenerative diseases in reasonable detail. The authors do assess the state of the literature with regards to potential therapeutic targets in relation to neurodegenerative disorders and associated cognitive deficits and they discuss papers that demonstrate some of the points they are making.
There could be one or two more supporting diagrams in this review that would help the reader and would highlight areas of importance in a concise and precise way.
The review itself covers many relevant previous publications and it does attempt to emphasise the limitations of some as well as highlighting the usefulness of others. It does cover what it says in the title.
Line 145: I think this should begin with “One quarter of PD patients…”(rather than One-fourth..). Figure 1 would appear to have red and green arrows missing in order to highlight where there is up-regulation or down-regulation of the cognitive deficits? The box at the bottom showing red and green arrows does not make sense. What do the red and green arrows actually mean in the context of Figure 1? Where is Figure 1 actually cited from as the text states reference 23 (line 91) but the Figure only mentions reference 58. Can the authors please rephrase their sentence on line 185 that currently states “……that ends up with death”. This would be better if it was more scientific in context. The whole manuscript is quite difficult to read and I think this is partly due to a lack of proof-reading. There are some interesting points being made but they are sadly lost in amongst the slightly awkward English language and the typing errors along with some minor grammatical issues. If these were rectified, the review would be much easier to follow and to digest. The title for Table 1 needs correcting (deferent - maybe this should say different?). I cannot find any mention of Figure 2 in the actual text. Can this please be rectified. There are numerous places throughout the text where the English is less than correct and so I suggest the manuscript is carefully checked by a native English speaker. I have listed some examples taken from pages 1 to 7 that can be found in the following places:Lines 20-21; 23-24; 26-28; 37; 40; 44; 45; 51; 54; 59; 63-64; 72; 73; 75-76; 81; 83; 84; 86-88; 88-90; 92; 97; 101-102; 106-107; 111; 112; 116-117;120; 121; 122-123; 124; 127-128; 130; 133-135; 137-138; 139-140; 143-144; 147-148; 148-149; 153-154; 162; 163; 164-165; 165-166; 166; 168; 171; 184; 186; 186-188; 200-202; 205; 206; 207; 208; 211; 214; 215-216; 216; 218-219; 219-221; 225-226; 226-228; 228-230; 232-233; 235-237; 237-238; 243; 244-245; 245; 247-248; 259; 265.
The document requires basic proof-reading throughout in order to eliminate the slightly annoying typographical errors that keep popping up. If these were corrected, the manuscript would be far more pleasant to read and it would make more scientific sense.
Author Response
All the question raised were of greater improvement of our manuscript, therefore we appreciate and tried our best to address all of them. In the word file affixed herewith contains reply. We hope we addressed all comments properly and if not please allow us to improve further.

Reviewer 2 Report
The manuscript reviews the role of G-protein coupled receptors in the pathogenesis of neurodegenerative disorders (NDDs), and provides potential therapeutic target for NDDs. However, it is not well organized and quite confusing. I have the following concerns:
In part 2, the authors talked about several GPRCs relative to NDDs. I suggest a clear description of GPCRs classification and function before that, so the readers would know why and how these receptors are involved. It is quite confusing that part 2 is not consistent with part 3. For example, the authors emphasized the importance of dopamine receptors in PD, however, in part 3 related to dopamine receptors, the authors talked more about FTD. The same as mGluRs and HD, cannabinoid receptors and HD, etc. It would be better to add more elements in figure 2 for distinct conditions. It is better to involve GABAR in part 3 as they are mentioned in part 2. Page 3 line 44, have "been" affected. Page 15 line 581, considering oGPCRs "as" the novel target. A2AR is not clarified.Author Response
All the question raised were of greater improvement of our manuscript, therefore we appreciate and tried our best to address all of them. In the word file affixed herewith contains reply. We hope we addressed all comments properly and if not please allow us to improve further.

Round 2
Reviewer 1 Report
In this revised version of the manuscript, the authors have clarified a proportion of the main concerns raised by the Reviewers and they have commented on some of the points made although they have not necessarily made any evident changes to the text itself. In some instances they have given their own views but have not made any changes to go with them.
For example: The authors response to one point: “We appreciate the concern raised. In part 2 we focused on the disease pathophysiology whereas in part 3 we focused on GPCRs family. Therefore, in part 2 we talked more about the disease and relevant one or multiple GPCRs. In part 3 where we tried to discuss more about current approaches with different GPCRs, we have correlated one or multiple disease conditions with a single GPCR family. We hope reviewer would understood that each sections are correlated with each other and as the first section is about single disease, so, the later part would seems a little different. But literally both parts are correlated with each other.”
I am personally not quite sure what that means and it doesn’t appear that the authors have actually done anything to rectify the issue for one of the Reviewers.
- There has been a change in the title of the Review that makes sense and I think this better describes the review and its content.
- I feel the authors gave quite a short reply to the comments made by the Reviewers but they have obviously tried to make some changes to the manuscript. I am not thoroughly convinced that they have addressed all the points raised (see the example above), but they have certainly made an attempt to rectify some of the issues.
- There appear to be 10 more references than in the previous version.
- I am slightly disappointed that no more diagrams were added but the review is already quite long and so this is probably for the best.
- The authors have made a serious attempt to rectify the majority of the English language issues and so the manuscript is much easier to read now.
- As I said in the initial report, the Review does cover some good science and it assesses the main neurodegenerative diseases in reasonable detail. The authors do assess the state of the literature with regards to potential therapeutic targets in relation to neurodegenerative disorders and associated cognitive deficits and they highlight papers that demonstrate some of the points they are making.
- I still think there is some confusion regarding parts 2 and 3 of the Review. The authors comment in their reply is actually more confusing than the text itself.
- Overall, I feel there has been some improvement to the whole manuscript.
Reviewer 2 Report
The authors answered part of my concerns. They did make an effort to revise the manuscript.